# Autologous Haematopoietic Stem Cell Transplantation and Systemic Sclerosis: Focus on Interstitial Lung Disease

**DOI:** 10.3390/cells11050843

**Published:** 2022-03-01

**Authors:** Gianluca Bagnato, Antonio Giovanni Versace, Daniela La Rosa, Alberta De Gaetano, Egidio Imbalzano, Marianna Chiappalone, Carmelo Ioppolo, William Neal Roberts, Alessandra Bitto, Natasha Irrera, Alessandro Allegra, Giovanni Pioggia, Sebastiano Gangemi

**Affiliations:** 1Department of Clinical and Experimental Medicine, University of Messina, 98125 Messina, Italy; gianbagnato@gmail.com (G.B.); agversace@unime.it (A.G.V.); albi.degaetano@gmail.com (A.D.G.); egidio.imbalzano@unime.it (E.I.); marianna.chiappalone@gmail.com (M.C.); m.iop91@gmail.com (C.I.); abitto@unime.it (A.B.); nirrera@unime.it (N.I.); alessandro.allegra@unime.it (A.A.); gangemis@unime.it (S.G.); 2Department of Medicine, University of Kentucky, Lexington, KY 40506, USA; neal.roberts@uky.edu; 3Institute for Biomedical Research and Innovation, National Research Council of Italy, 98125 Messina, Italy; giovanni.pioggia@irib.cnr.it

**Keywords:** systemic sclerosis, interstitial lung disease, hematopoietic stem cells transplantation

## Abstract

Autologous hematopoietic stem cells transplantation (AHSCT) has been employed as treatment for severe systemic sclerosis (SSc) with high risk of organ failure. In the last 25 years overall survival and treatment-related mortality have improved, in accordance with a better patient selection and mobilization and conditioning protocols. This review analyzes the evidence from the last 5 years for AHSCT-treated SSc patients, considering in particular the outcomes related to interstitial lung disease. There are increasing data supporting the use of AHSCT in selected patients with rapidly progressive SSc. However, some unmet needs remain, such as an accurate patient selection, pre-transplantation analysis to identify subclinical conditions precluding the transplantation, and the alternatives for post-transplant ILD recurrence.

## 1. Introduction

Systemic sclerosis (SSc) is a heterogeneous, systemic autoimmune disease characterized by small vessel vasculopathy, autoantibodies production, and fibroblast activation leading to fibrosis of the skin and internal organs [1]. The clinical manifestations and the prognosis of SSc are variable: the majority of patients have skin thickening and variable involvement of internal organs. Among the different forms of SSc, classified according to the extent of skin thickening [2], patients with the diffuse form (dcSSc) show earlier and more frequent organ involvement [3]. The most relevant complications for SSc are related to pulmonary involvement and these are the direct expression of the pathogenic features of the disease: pulmonary arterial hypertension (PAH), as the most aggressive manifestation of vascular damage, and interstitial lung disease (ILD), as the most life-threatening manifestation of the fibrotic process of SSc [4]. Among these, SSc-associated interstitial lung disease (SSc-ILD) represents the leading cause of mortality in patients with SSc with 3-year survival of only 52% [5]. The clinical course of SSc-ILD is variable: some patients show stable or improving forced vital capacity (FVC) while others show a progressive decline in lung function [6], that occurs more rapidly within the first few years after diagnosis and then in some cases slows down. The clinical heterogeneity of the disease in association with the poor survival rate makes clinical trial enrichment difficult and complicates stratification and therapeutic management [7].

Over the past 20 years, cyclophosphamide (CYC) has been considered the standard of care for SSc with early aggressive skin disease and concomitant progressive ILD [8,9], and more recently, mycophenolate mofetil (MMF) has been added to the therapeutic options for the management of SSc-ILD, after the results of the Scleroderma Lung Study II [10]. In addition, the most recent recommendations of the panel of experts support the use of hematopoietic stem cell transplantation (HSCT) in SSc patients with rapidly progressive disease and high risk of organ failure based on two randomized clinical trials (RCTs) that evaluated the efficacy and safety of high-dose immunosuppressive therapy with subsequent HSCT [11,12]. Indeed, these trials showed that HSCT improves event-free survival compared to CYC, an endpoint that is not reached by any pharmacological treatment to date. Nonetheless, the clinical decision to address SSc patients to HSCT is further complicated by the identification of essential requirements: knowing with a degree of certainty that the disease is rapidly progressive, which depends also on the medications currently used by the patient; convincing the patient, which might be challenging since these are SSc patients at high risk of organ failure that do not realize the actual risk of their disease; assessing that the disease is progressing in a monotonic fashion without chances to slow down spontaneously to identify the appropriate timing.

As a form of intensive immunotherapy that targets the autoreactive adaptive immune system, HSCT deeply modifies the immune system, restoring the immunological balance that counteracts inflammation and fibrosis progression, enabling disease control and, eventually, tissue repair [13]. Briefly, HSCT consists of four steps: mobilization of hematopoietic stem cells using chemotherapy, such as CYC, and growth factors [granulocyte colony-stimulating factor (G-CSF)]; conditioning using myeloablative or non-myeloablative regimens to eradicate autoreactive immune cells; reinfusion of autologous stem cells and immune reconstitution [14]. In contrast to myeloablative stem cell transplant, non-myeloablative transplant differs primarily in what happens prior to the transplant, since it employs much lower and less toxic doses of chemotherapy. Different combinations of mobilization and conditioning regimens have been used in randomized clinical trials in SSc-ILD.

The aim of the present review is to analyze the evidence from observational and retrospective studies and RCTs regarding autologous hematopoietic stem cells transplantation (AHSCT) in SSc-ILD published between January 2016 and December 2021. Studies involving less than 10 patients were excluded, as well as study designs, animal studies, case reports, review articles, letters to the editor, conference abstracts, editorials, and guidelines.

## 2. HSCT and SSc-ILD

### 2.1. Efficacy of HSCT in SSc-ILD

#### 2.1.1. Randomized Clinical Trials

Two randomized controlled trials, ASSIST (American Scleroderma Stem cell versus Immune Suppression Trial) in 2011 and ASTIS (Autologous Stem cell Transplantation International Scleroderma trial) in 2014 [12], both using non-myeloablative regimen, demonstrated the superiority of AHSCT compared to CYC. In these two trials the conditioning phase consisted of a non-myeloablative regimen with CYC (200 mg/kg total dose) and rabbit antithymocyte globulin (rATG). The main differences were the CYC dose, and the selection of peripheral blood stem cells used for the mobilization phase.

In the ASSIST trial, 2 g/m^2^ CYC was administered, and unmanipulated peripheral blood stem cells were infused, while in the ASTIS trial stem cells were mobilized with 4 g/m^2^ CYC and CD34+ selected peripheral blood stem cells were infused. Both trials used similar inclusion criteria (patients age between 18 and 65 years, with dcSSc and a maximum disease duration of 4 years, minimum modified Rodnan skin score (mRSS) of 15 and involvement of heart, lungs, or kidney). The ASSIST trial showed that non-myeloablative AHSCT improves skin and pulmonary function for up to 2 years. In the ASTIS trial the superiority of AHSCT vs. CYC was confirmed. Despite the occurrence of early treatment-related mortality during the first year after AHSCT (16.5%), and an increase in serious adverse events, especially characterized by respiratory and cardiac failure, the use of AHSCT was associated with increased long-term event-free survival [12].

In 2018 an additional randomized clinical trial, the Scleroderma: Cyclophosphamide or Transplantation trial (SCOT) was published. In this trial similar inclusion criteria were used, with GCS-F for mobilization and with CD34+ selected cells. The myeloablative regimen employed conditioning using a lower dose of CYC (120 mg/kg total dose) plus total body irradiation (TBI) (800 cGy/4 fractions over 2 days/200 cGy to lungs and kidneys with shielding) and equine ATG [15]. The trial enrolled 75 patients with severe SSc. Thirty-four were assigned to HSCT and 39 to CYC, ending at month 54, with a maximum follow-up to 72 months. One of the main differences, in comparison to previous RCTs, is the selection of the primary endpoint. This was identified in the global rank composite score (GRCS), an analytic tool which includes multiple disease features, including death and event-free survival, FVC, the Disability Index of the Health Assessment Questionnaire (HAQ-DI) score, and the mRSS. After 54 months, a large percentage (67%) of patients, treated with HSCT, reached the primary endpoint, the improvement of GRCS when compared with CYC arm (33%). Of note, the exclusion criteria were rigorous. These included active gastric antral vascular ectasia, a diffusing capacity of the lung for carbon monoxide (DLco) of less than 40% of the predicted value, an FVC of less than 45% of the predicted value, a left ventricular ejection fraction of less than 50%, a creatinine clearance of less than 40 mL per minute, pulmonary arterial hypertension, or more than 6 months of previous treatment with CYC. Treatment-related mortality, in the transplantation group, was 3% at 54 months and 6% at 72 months, when compared with 0% in the CYC group. However, transplant-related mortality was lower than previous reports, but the reason is still unknown, and it might be partly explained by the differences in inclusion criteria (in the SCOT trial none of the patients had cardiac involvement) and in conditioning regimens (in the ASTIS trial high-dose regimen of CYC may has been toxic, especially in the presence of heart disease).

Notably, in this trial, treatment of severe scleroderma with myeloablative therapy and CD34+ selected autologous hematopoietic stem-cell transplantation led to superior long-term outcomes as compared with standard therapy and fewer scleroderma relapses, defined as the need for disease-modifying antirheumatic drugs (DMARD) therapy after non-myeloablative regimen, probably due to T cell depletion after total body irradiation. However, the adverse events occurring in the first 2 years after transplantation, such as viral infections and secondary cancers due to total body irradiation exposure, were relevant (96% in the transplant group vs. 71% in the CYC group). Cancers occurred in four participants: three in the transplantation group (one had papillary thyroid cancer and two had the myelodysplastic syndrome) and one in the CYC group (breast cancer). A total of 21 deaths occurred over a period of 72 months in this trial, 14 in the CYC group and 7 in the transplant group.

#### 2.1.2. Prospective and Cross-Sectional Studies 

Prospective studies in the last 5 years have shown that AHSCT in systemic sclerosis resulted in long-term improvements in skin thickness and pulmonary function, but its efficacy may be limited by the relatively high treatment-related mortality, and morbidity. Patients with dcSSc for less than 2 years, without severe cardiac and renal involvement, without pulmonary hypertension, exposed to non-myeloablative conditioning regimen with high-dose of CYC and CD34+ selected cells, benefit from AHSCT with improvement in skin thickness and improvement or stability of pulmonary function [14,15,16,17]. A post hoc analysis of a phase I/II clinical trial conducted in Japan compared the efficacy and safety of CD34-selected auto-HSCT in comparison to unmanipulated auto-HSCT. The authors observed that FVC improved continuously for 8 years in patients that underwent CD-34 selection before transplant, while in the unmanipulated group FVC returned to baseline after 3 years. This was accompanied by an increase in toxicity and in viral infections rate in the CD-34-selected group without differences in serious adverse events or death [16]. In addition, a large multicenter non-interventional study analyzed real-life efficacy of AHSCT including 80 SSc patients using as primary endpoint progression-free survival (PFS), defined as survival after AHSCT without death or evidence of progression of SSc [15]. While, in this study, the mobilization regimen was similar across the centers (CYC 1–4 g/m^2^) and the conditioning process was not based on myeloablative approaches, the main difference in terms of efficacy was observed in those exposed to CD-34 selection with a better clinical response. Interestingly, long term data from a Japanese phase II trial, with a median follow-up period of 137 months involving 14 patients, confirmed a high overall survival (93%) and a stable pulmonary function in 85% of the subjects treated with HSCT [16]. Several authors have studied patient serum cytokine profiles before and after AHSCT to assess whether immune monitoring can be useful in predicting clinical response [17,18,19] A modified immune response is essential to SSc physiopathology, and an alteration in the cytokines and growth factors seems to precede tissue fibrosis. High levels of IL-2, IL-6, IL-8, MCP-1, and IFN-γ reported in SSc patients before AHSCT confirm the pro-inflammatory features of the disease. After AHSCT, authors observed a significant reduction in IL-2 and IL-8 levels, and a slight but significant decrease in TGF-β which may reflect a decrease of SSc fibroblast activation after AHSCT in dcSSc subjects. Monocyte chemoattractant protein 1 (MCP-1) also participates to fibrosis via different mechanisms, comprising augmented fibroblast growth and collagen production, and MCP-1 amounts decreased after transplant with progressive reduction over the 48 months after transplant. Finally, platelet derived growth factor (PDGF) is linked to fibrosis, and although the study found no augmentation in circulating levels of PDGF before transplant, a significant reduction at 6 months after AHSCT was reported [20], (Table 1).

Clinical improvement of SSc patients may be related to increased counts of newly generated Tregs and B regs after AHSCT as a result of coordinated thymic and bone marrow rebound [18]. High PD-1 expression is associated with better clinical outcomes after AHSCT [17]. Antibody reactivity towards topoisomerase-I decreases after HSCT and correlates with good clinical response while the reactivity to the epitope anti-topo489–573 was significantly higher in SSc patients that did not respond to AHSCT as compared with subjects with a good response [19].

#### 2.1.3. Retrospective Studies 

In a retrospective study 41 SSc patients undergoing HSCT were compared with 65 SSc patients treated with conventional care. Patients fulfilled the HSCT eligibility criteria of the ASTIS trial (age > 18 years, dcSSc, maximum disease duration of 4 years, minimum mRSS of 15, lung, heart, and renal involvement). The HSCT resulted in a reduction of the mRSS and a stability of the DLco. Pre-transplant cardiac screening is therefore essential not only for risk assessment during the treatment but also to anticipate response [25]. Long-term retrospective analysis of HSCT among all patients treated with HSCT for SSc in the Netherlands until 2017 (*n* = 92) gave estimates of event-free survival at 5, 10, and 15 years as 78%, 76%, and 66%, respectively. In this cohort, FVC increased during follow-up from 84% to 94% at 5 years of follow-up, while DLco increased from 55% to 61% at 5 years, which is significant but not clinically meaningful [26]. Similarly, Del Papa et al. demonstrated, in a head-to head comparison between HSCT and CYC study, that both FVC and DLco remain unchanged after AHSCT [27]. Indeed, other retrospective studies confirm that both Dlco and FVC improve after HSCT [25,28]. Furthermore, an improvement of clinical scores within the first year after transplant is a predictor for long-term clinical responses [24], (Table 2).

### 2.2. HSCT Safety and Complications in SSc-ILD

A crucial point for AHSCT in systemic sclerosis is safety. The risk of early transplant-related complications and mortality varies according to the type of disease, pre-existing internal organ involvement, the experience of the transplant center, and the intensity of the conditioning regimen [30]. Major adverse events and treatment-related mortality (TRM) after HSCT are predominantly related to primary cardiac and lung involvement [11,12,15]. They represent the leading cause of mortality after AHSCT for autoimmune disease and cluster within the first month after AHSCT. While in the ASTIS trial, viral infections were detected in 27.8% of patients in the group of SSc patients treated with AHSTC group versus 1.3% in the control arm, overall infection rates were similar in the two treatment arms of the SCOT trial except for varicella zoster infection that was detected in 36% of AHSCT treated patients. TBI used in myeloablative conditioning regimen is more lethal for hematopoietic stem cells. It is responsible for the hematological neoplasms, as observed in the comparison between ASTIS and SCOT trials and might account for the cardiopulmonary deaths as observed in patients undergoing myeloablative regimen [15,16,31,32,33]. High-dose cyclophosphamide, however, included in myeloablative conditioning regimens, is also toxic for the heart function, leading to life-threatening myocardial hemorrhage, fatal cardiomyopathy, and heart failure [34]. The specific effect of TBI and CYC on cardiac function needs to be better clarified to assess the contribution of each component to treatment-related outcomes. Based on the results presented by Helbig G. et al. [22], the use of alemtuzumab, in association with CYC, for the conditioning phase, was associated with an increased occurrence of infections and thus it should be avoided because of high risk of developing infectious complications. Indeed, one of the main clinical unmet needs is the identification of the most effective conditioning protocol avoiding the toxicity induced by over-treatment and the concomitant increase in infection rate. The ASTIS trial showed that the administration of CYC at 200 mg/kg body weight with rATG 7.5 mg/kg body weight seems to be the best protocol [12].

Many factors influence the outcome of transplant, and they are as follows: HSCT protocol, age of patient, disease duration before transplant, quality of life, and co-morbidities, e.g., decreased left ventricle ejection fraction (LVEF) or pulmonary arterial hypertension (PAH). The most frequent complications are viral reactivations (CMV, EBV, HSV) and opportunistic infections [21,22,27,29]. Reactivation may be dependent on ATG in conditioning [29] and occurs more frequently in CD34-selected patients [21] and in patients with low B cell count prior to transplant [29]. Determination of the CMV and the EBV serostatus before HSCT seems advisable to identify patients at risk for virus reactivations and prophylaxis may be useful in patients with positive serology. HSCT allows patients with refractory autoimmune disease during childbearing years the opportunity to conceive during treatment-free remissions with favorable pregnancy outcomes and reduces the risks of maternal complications of the underlying disease [35].

### 2.3. SSc-ILD Patients Selection for HSCT

Cardiac dysfunctions that occur during transplantation accounted for many of the transplant-related deaths described in several studies [15,16,23,26]. Male sex, older age, and LVEF < 50% are independently associated with events [26], as is smoking as reported in the ASTIS and SCOT trials [12,15]. A relevant decrease of mortality rate could be achieved with a more accurate selection of candidate patients and a better pre-transplant definition of cardio-pulmonary risk. Cardiac evaluation before use of mobilization and conditioning regimen includes electrocardiogram and echocardiography with pulmonary artery pressure to exclude patients with pulmonary arterial hypertension [30]. However, the screening of cardiac function using only electrocardiogram and echocardiography remains insufficient in diagnosing occult heart disease [36]. Since 2012 this evaluation was extended with left and right heart catheterization including fluid challenge, 24 h Holter registration if history of palpitations or abnormal ECG, and cardiac MRI when indicated [37].

## 3. Discussion

In this review we summarized the results of studies of the last 5 years employing AHSCT for the treatment of SSc patients at high risk of organ failure. AHSCT ameliorates skin involvement and improves or stabilizes lung function, especially in patients with disease duration of less than five years [13,38]. The patient-related perspective confirms that this procedure is well-tolerated and improves quality of life, as shown by a recent systematic review [39]. Robust evidence for the efficacy of AHSCT in SSc-ILD is, however, still lacking because HSCT trials and studies have differences in inclusion criteria, treatment regimens, and clinical endpoints and a comparison between them remains difficult. Although this procedure maintains a 5–15% risk of mortality, it induces, in the great majority of treated patients, improvement in survival, disease activity and skin thickening, and, eventually, a stabilization of lung function. However, it is unclear how far forward AHSCT should be positioned as an upfront therapy vs. its role as a rescue treatment for patients’ refractory to immunosuppressive therapy. As immunosuppressive and antifibrotic therapies advance in efficacy and specificity, the opportunities to apply AHSCT as a rescue therapy recede but as its safety improves, the opportunities to use AHSCT earlier increase. Overall, early toxicity remains the principal challenge. In particular, a more individualized regimen with respect to heart, lung or kidney involvement might further improve patients’ safety by reducing toxicity and treatment-related mortality of the conditioning phase.

Many centers select the cells infused into the patients by removing T cells and concentrating CD34+ cells. This approach has controversial results and thus the benefits of ex vivo CD34+ positive selection of the graft remains debated. A potential beneficial effect might be the preliminary elimination of auto-reactive cells, even if a recent multicenter retrospective study demonstrated that the use of selected CD34+ cells in SSc confers no benefit over unselected cells [40]. CD34+ selection may delay T cell reconstitution for the first 8 months after transplant [41] and remove auto-reactive cells and also regulatory T-cells. Indeed, post-transplant CD4 T-cell reconstitution correlates with long-term clinical response post-transplant [17]. Further studies are still required to explore the benefits of CD34+ selection. On the other hand, changes in the conditioning regimen among the trials provided different results in terms of efficacy and safety. The SCOT trial, employing low dose CYC and TBI, induced three times more malignancies (9%; two cases of myelodysplastic syndrome and one of medullary thyroid cancer). Indeed, the use of myeloablative conditioning regimen is the main difference in this trial compared to previous trials and must be accounted for when comparing the results of these trials. However, the SCOT trial reported an impressive treatment-related mortality of 3% at 54 months and 6% at 72 months, superior to previous trials. There are several explanations for these findings: first, the use of lower dose CYC, has less cardiotoxic effects as reported in the adverse events profile of the SCOT trial. Furthermore, the exclusion of smokers and patients with cardiac dysfunction in association with a better cardiac function assessment by using right heart catheterization with fluid challenge and cardiac magnetic resonance imaging to identify early cardiac involvement might have reduced the number of SSc at high risk of treatment-related complications.

Although new pharmacological therapies are emerging, international guidelines recommend treating selected patients with early rapidly progressive dcSSc with AHSCT in highly experienced bone marrow transplant centers. Notably, in recent years new pharmacological therapies have been tested in phase III randomized clinical trials for SSc-ILD, increasing the options for the management of this disease. The phase III trials of nintedanib [42] and tocilizumab [43] for SSc-ILD patients represent an advancement in the management of SSc-ILD. However, overall survival is not included in the primary endpoints of these trials and this endpoint is the most relevant aspect when comparing pharmacological and HSCT trials in SSc. On the other hand, in trials using AHSCT ILD presence per se is not used as a single inclusion criterion, thus the comparison with pharmacological trials becomes even more problematic [40]. A recent retrospective real-world study aimed at assessing the evolution of ILD in SSc patients treated with AHSCT and CYC at 1 year. This study show that 90% of patients treated with HSCT or CYC present stable or reduced ILD extent assessed by HRCT, although no definitive conclusion can be drawn for the superior efficacy of one regimen upon the other, possibly due to the relative short period of observation [41].

The comparison between the SCOT trial and the previous trials, despite having similar eligibility criteria and control treatment, is limited by the variability of the endpoints. In the SCOT trial the use of a composite endpoint, the global rank composite score (GRCS), provides a systemic view of the disease manifestations and treatment effect. However, this approach limits the analysis of the results: first, this score has not yet been validated and thus the individual components require an accurate sub-analysis to identify the within-score differences. In addition, confirmation of their validity is needed before they can achieve widespread acceptance and, in the case of SSc trials, they have never been used and validated. Interestingly, Farge et al. applied the GRCS to the French participants of the ASTIS trial to compare the efficacy according to the same endpoint used in the SCOT trial [41], supporting its use to standardize the endpoint for future trials.

This is particularly important when analysing the efficacy of AHSCT in the treatment of ILD. The comparison between the results of the trials and other retrospective and prospective studies suggest that a clear definition of the appropriate pulmonary function tests to use as a marker of disease progression needs to be better clarified and might be related to baseline characteristics of patients treated with AHSCT [44]. In particular, DLco does not improve in patients with cardiac involvement before transplant but only in those with normal cardiac tests at baseline. In fact, longer-term follow-up of pulmonary function and alterations shows that the effects of AHSCT on the evolution of ILD tend to decrease over-time and eventually disappear in some patients, returning to the pre-transplant conditions two years after AHSCT. For this group of patients, Rituximab, an anti-cd20 biologic agent, showed a promising profile of safety and efficacy, offering the opportunity to avoid ILD recurrence or worsening after AHSCT [45]. This directly raises the question whether future trials should focus on the stabilization of pulmonary function improvement after AHSCT and to identify SSc patients at risk of long-term ILD progression or re-activation.

SSc is characterized by auto-antibodies production and T-cell infiltration. The various cytokines and chemokines produced by the activated T cells recruit additional inflammatory cells and activate fibroblasts, that in turn promote extracellular matrix production and excessive fibrosis within the skin and internal organs. The exact mechanism of action of HSCT remains unknown. Some authors hypothesize that AHSCT resets the immune system with re-emergence of naive B cells and T cells and progressive increase in the regulatory T-cell number and in the Th1/Th2 cell ratio [46,47]. This might lead to the occurrence of secondary autoimmune disorders and engraftment syndrome that is much more frequent in patients with cardiac involvement, thus further reinforcing the relevance of an accurate screening of cardiac subclinical alterations and the opportunity to employ less cardiotoxic approaches, such as fludarabine [48,49].

## 4. Conclusions

Recent advances in SSc-ILD pathogenesis and emerging phase III clinical trials for new pharmacologic treatments for the treatment of SSc-ILD show convincing data regarding stabilization of pulmonary function, but no pharmacological trials have yet achieved increased overall survival as primary endpoint. The results of recent phase II and phase III trials, however, warrant a revision of the actual trial designs for AHSCT in order to include a better definition of the comparator drug. In addition, accumulating evidence from recent randomized clinical trials employing AHSCT suggest that the combination of a better cardiac evaluation of SSc patients, in association with CD-34+ selection in the mobilization phase and a reduced dose of CYC in the conditioning phase, confers a favorable long-term overall survival with an acceptable safety profile [14]. However, the progression of lung disease remains an unmet need and requires an accurate stratification of these patients to assess their eligibility for further pharmacological maintenance therapy with new emerging SSc-ILD specific drugs.

## 5. Future Directions

HSCT has been used for more than 20 years as a specific treatment in a wide range of autoimmune disease [50,51]. The rationale for HSCT is based on its capacity to reset the immune system after eradication of the autoreactive cells with high immunosuppressive or myeloablative conditioning regimen, allowing the reconstitution of immune tolerance. The UPSIDE trial is ongoing and investigates AHSCT in early disease compared to other immunosuppressive therapy with the specific aim to treat the immunosuppressive arm with AHSCT as a rescue therapy if the response is poor [36]. In SSc, the debate on the potential benefits and risks of transplantation and the correct identification of high-risk subjects remains difficult and the recent approval of new immunosuppressive and anti-fibrotic drugs offers the possibility to provide long-term beneficial effects for the management of ILD both before and after AHSCT [52]. Notwithstanding the new drug approvals and the ambiguities of protocol in its use, AHSCT remains the only intervention to offer proven survival benefit in SSc thus far.

## Figures and Tables

**Table 1 cells-11-00843-t001:** Prospective studies.

	SCOT Trial [15]	Ayano, M., et al. [21]	Helbig, G., et al. [22]	Henes, J., et al. [23]	Nakamura, H., et al. [16]	Farge, D., et al. [24]	Michel, L., et al. [20]	Arruda, L., et al. [17]
Participants	Total: 65 HSCT: 33 CYC: 32	Total: 19 CD34+: 11 CD34−: 8	Total: 18 CYC + alemtuzumab: 11 Melphalan + alemtuzumab: 2 CYC and rATG: 4 CYC alone: 1	Total: 80	Total: 14	Total: 10	Total: 38	Total: 31
Disease duration (years)	<4	<4	<10	<2	<3	<2	<2	<2
Age	18–69	16–65	18–70	18–65	16–60	18–65	18–65	19–58
PFTs	FVC: 45–70 DLco: 45–70	FVC: <70 DLco: <70	FVC: 40–80 DLco: 40–80	FVC: >40 DLco: >40	FVC: >45 DLco: >45	FVC: >50 DLco: >45	FVC: <70 DLco: 45–70	FVC: >45 DLco: >45
mRSS	>15	>15	>15	>15	>15	>15	>20	>14
Heart involvement	no	no	no	no	no	no	no	no
Follow-up (months)	54	96	42	24	144	104	48	36
Mobilization	G-CSF	G-CSF + CYC	G-CSF + CYC	G-CSF + CYC	G-CSF + CYC	G-CSF + CYC	G-CSF + CYC	G-CSF + CYC
Conditioning	TBI 800 cGy + CYC 120 mg/kg + eATG 90 mg/kg vs. CYC 750 mg/mq	CYC (50 mg/kg)	CYC 200 mg/kg + alemtuzumab 60 mg, or melphalan 140 mg/mq + alemtuzumab, or CYC + rATG 7.5 mg/kg, or CYC	CYC 200 mg/kg	CYC 200 mg/kg	CYC 50 mg/kg + ATG	CYC 50 mg/kg + ATG	Cy 200 mg/kg + 4.5 mg/kg ATG
CD-34 selection	yes	yes (11 patients)	no	yes (35 patients)	yes (5 patients)	yes (5 patients)	yes	no
Event-free survival (EFS) or Progression-free survival (PFS)	EFS: 74% in HSCT vs. 47% in CYC at 6 years	PFS: 51.3%	PFS: 33%	PFS: 81.8%	40% at 10 years	-	-	-
Overall survival	86% in HSCT vs. 51% in CYC at 6 years	79%	61%	90%	93% at 10 years	-	-	97%
Transplant-related mortality	3%	zero	24% (protocol amendment for alemtuzumab)	6%	7.1%	-	-	-
Pulmonary function outcomes	HSCT superior	FVC improved at 8 years- DLco stable	FVC and DLco stable at 12 months	FVC improved-DLco stable	85% stable at 12 months	-	-	-

**Table 2 cells-11-00843-t002:** Retrospective studies.

	Gernert, M., et al. [29]	Van Bijnen, S., et al. [26]	Del Papa, N., et al. [27]	Henrique-Neto et al. [28]
Participants	17	89	Total: 54 HSCT: 18 Standard care: 36	70
Disease duration (years)	<4	1.5 (median)	<4	Up to 7 years
Age	23–64	46 (median)	20–64	19–59
PFTs	FVC >58 DLco: >30	FVC: 85 (median) DLco: 45 (median)	FVC > 50 DLco > 50	FVC < 45% DLco > 40%
mRSS	>20	>14	>14	>14
Heart involvement	yes	yes	no	-
Follow-up (months)	12	60	60	60
Mobilization	G-CSF + CYC	G-CSF + CYC	G-CSF + CYC	G-CSF + CYC
Conditioning	CYC 200 mg/kg + ATG 30 mg/kg	CYC (50 mg/kg) + rATG 2.5 mg/kg + methylprednisolone 1 mg/kg	CYC 200 mg/kg + rATG 7.5 mg/kg	CYC 200 mg/kg + ATG 4.5 mg/kg (for patients with heart involvement: fludarabine 120 mg/m^2^ + melphalan 120 mg/m^2^ + 4.5 mg/kg ATG
CD-34 selection	yes	yes	yes	no
Event-free survival (EFS) or Progression-free survival (PFS)	-	EFS: 78% at 5 years	-	PFS: 70.5% at 8 years
Overall survival	-	77% at 55 months	89% in SHCT vs. 39% in CYC at 5 years	81% at 8 years
Transplant-related mortality	5.9%	11%	5.6%	4%
Pulmonary function outcomes	-	FVC and DLco improved at 5 years	Stabilization of FVC and DLCO in HSCT. No difference between HSCT and CYC	FVC and DLco stable at 5 years

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
