# Peer review of "Autologous Haematopoietic Stem Cell Transplantation and Systemic Sclerosis: Focus on Interstitial Lung Disease"

_cells, 2022, doi:10.3390/cells11050843_

Round 1
Reviewer 1 Report
The authors present the state of the art in the use of AHSCT for Systemic Sclerosis with a specific fucus on ILD. This review presents the main clinical studies on AHSCT in a clear way also underlining the risks and benefits and the difficulty of comparing this type of therapeutic regimen with other drugs currently used. The authors also point out the unmet needs that make it difficult to conduct clinical trials in this field. The review is well conducted and covers a very important topic but I suggest to:
-Clarify/correct the definition of non myeloablative transplant (row 74-76). The sentence states that this kind of transplant is "followed by the infusion of donor stem cells instead of the patient stem cells". However, the article mentions autologous transplantation, i.e., from the patient, and the same clinical trials cited use autologous transplantation.
-Correct some typos in the text:
- Row 62: there is a comma instead of a period;
- Row 76: check grammar
- Row 190: check the sentence
- Row 212: the acronym TBI appears which is not used near the extended definition (total body irradiation row 100 and row 205)
Author Response
We thank the reviewer for the relevant comments.
1Q- Clarify/correct the definition of non myeloablative transplant (row 74-76). The sentence states that this kind of transplant is "followed by the infusion of donor stem cells instead of the patient stem cells". However, the article mentions autologous transplantation, i.e., from the patient, and the same clinical trials cited use autologous transplantation.
1A We modified the paragraph according to the suggestion in row 74-76
2Q-Correct some typos in the text:
- Row 62: there is a comma instead of a period;
- Row 76: check grammar
- Row 190: check the sentence
- Row 212: the acronym TBI appears which is not used near the extended definition (total body irradiation row 100 and row 205)
2A The typos have been corrected
- row 62 period inserted, comma deleted
- row 76 the phrase has been modified
- row 190 the sentence has been corrected
- row 212 the acronym TBi was inserted when it first appears in the text and used accordingly throughout the body of the manuscript
Reviewer 2 Report
As authors state, the present review analyzes the evidences of autologous hematopoietic stem cell transplantation in Systemic Sclerosis. Authors review the last 5 years published papers.
The review is well structured and written and it is easy to follow.
I have some criticism and suggestions.
- The main criticism is that authors should describe the method that they follow for searching the publishing papers in the field of AHSCT in SSc. They should also provide a chart flow or describe inclusion/exclusion criteria for review.
- Authors should provide a short rationale for AHSCT in SSc. They mention the rationale in the "Future directions " section but I think it should be comment on introduction for a clear understanding of readers.
- Authors mention in line 74 a comment regarding differences between myeloablative and non-myeloablative conditioning regimen but they state that "followed by infusion of donor stem cells instead of patients stem cells". Infusion of donor stem cells means "allogeneic stem cell transplantation". Please revise the mentioned statement.
- Authors mention that UPSIDE trial is ongoing in "Future directions" section but if I am not wrong the mentioned trial has been already published (see, Springs J et al. BMI Open 2021 Mar 18, 11(3):e044483.doi:10.1136/bmjopen-2020-044483.). Please provide an appropriate comment on this and the mentioned reference.
Author Response
We thank the reviewer for the relevant comments provided during this round of reviewer. Please find below the answers:
1Q The main criticism is that authors should describe the method that they follow for searching the publishing papers in the field of AHSCT in SSc. They should also provide a chart flow or describe inclusion/exclusion criteria for review.
1A We thank the reviewer for the relevant comment regarding the search method. As specified in the type of article in the front page, we did not mean to perform a systematic review mainly due to the absence of an appropriate amount of randomized clinical trials published in the last 5 years. Indeed, one of the main aspects regarding AHSCT in SSc is the high heterogeneity in the methods employed among the different studies. In fact, the aim of the actual review is to analyze the results of the most relevant studies published in the last 5 years, by selecting randomized clinical trials, retrospective, prospective and cross-sectional studies with more than 10 SSc patients to provide an analysis of the most robust and recent evidence regarding AHSCT in SSc with a specific focus on ILD. The manuscript has been adapted accordingly to your relevant suggestion.
2Q: Authors should provide a short rationale for AHSCT in SSc. They mention the rationale in the "Future directions " section but I think it should be comment on introduction for a clear understanding of readers.
2A: According to the suggestion of the reviewer, we added in the introduction additional elements regarding HSCT.
3Q Authors mention in line 74 a comment regarding differences between myeloablative and non-myeloablative conditioning regimen but they state that "followed by infusion of donor stem cells instead of patients stem cells". Infusion of donor stem cells means "allogeneic stem cell transplantation". Please revise the mentioned statement.
3A: the statement has been revised
4Q: Authors mention that UPSIDE trial is ongoing in "Future directions" section but if I am not wrong the mentioned trial has been already published (see, Springs J et al. BMI Open 2021 Mar 18, 11(3):e044483.doi:10.1136/bmjopen-2020-044483.). Please provide an appropriate comment on this and the mentioned reference.
4A: We verified again the reference and, as specified in the text, we confirm that this is the publication of the study protocol and not the final study. No results have been published so far since the study is still actively recruiting as reported in the clinicaltrial.gov page. The last update of the study status was provided on Dec 8th 2021 as reported in https://clinicaltrials.gov/ct2/show/NCT04464434.